
# The precision of satellite-based irrigation quantification in the Indus and Ganges basins

Søren J. Kragh[1], Rasmus Fensholt[2], Simon Stisen[1], Julian Koch[1]

[1]Department of Hydrology, Geological Survey of Denmark and Greenland, Copenhagen, 1350, Denmark
5   [2]Department of Geosciences and Natural Resource Management, University of Copenhagen, Copenhagen, 1350, Denmark

*Correspondence to*: Søren Julsgaard Kragh (sjk@geus.dk)

**Abstract.** Even though irrigation is the largest direct anthropogenic interference with the terrestrial water cycle, limited
10   knowledge on the amount of water applied for irrigation exist. Quantification of irrigation via evapotranspiration (ET) or soil
moisture residuals between remote sensing models and hydrological models, with the latter acting as baselines of natural
conditions without the influence of irrigation, have successfully been applied in various regions. Here, we implement an novel
ensemble methodology to estimate the precision of ET-based net irrigation quantification by combining different ET and
precipitation products in the Indus and Ganges basins. A multi-model calibration of 15 models independently calibrated to
15   simulate natural rainfed ET was conducted prior to the irrigation quantification. Based on the ensemble average, the 2003-
2013 net irrigation amounts to 246 mm/year (78 km$^3$/year) and 115 mm/year (76 km$^3$/year) in Indus and Ganges basin,
respectively. Net irrigation in Indus basin is evenly split between dry and wet period, whereas 73% of net irrigation occurs
during the dry period in Ganges basin. We found that although annual ET from remote sensing models varied by 91.5 mm/year,
net irrigation precision was within 25 mm/season during the dry period, which emphasizes the robustness the applied multi-
model calibration approach. Net irrigation variance was found to decrease as ET uncertainty decreased, which related to the
climatic conditions, i.e. high uncertainty under arid conditions. A variance decomposition analysis showed that ET uncertainty
accounted for 81% of the overall net irrigation variance and that the influence of precipitation uncertainty was seasonally
dependent, i.e. with an increase during the monsoon season. The results underline the robustness of the framework to support
large scale sustainable water resource management of irrigated land.

## 1 Introduction

Today, 40% of global irrigated cropland are sustained by groundwater abstraction (Siebert et al., 2010), which have made
regional groundwater levels decline as abstraction rates have exceeded the annual recharge (Malakar et al., 2021; Mujumdar,
2013; Rodell et al., 2009; Shekhar et al., 2020). By 2050, global food production will have to increase by 60 percent to meet
global food demand and 90% of this increase in food production is projected to take place in developing countries
(Alexandratos and Bruinsma, 2012). Water scarcity is thus likely to intensify and threaten the livelihood of hundreds of
millions people living in the affected areas as well as global food security (Jain et al., 2021; Mujumdar, 2013).



Despite this, our knowledge on the extent of irrigated areas and irrigated water use are limited. In recent years, mapping of irrigated areas from microwave or/and optical satellite data have advanced (Bazzi et al., 2021; Dari et al., 2021; Lawston et al., 2017; Sharma et al., 2021) and scientific advances have aimed at estimating irrigation water use by isolating
satellite-based ET or soil moisture as a non-precipitation source (Brocca et al., 2018; Jalilvand et al., 2019; Koch et al., 2020; Zaussinger et al., 2019; Zohaib and Choi, 2020). Knowledge on irrigated water use is important for correctly modeling of the water balance (Shah et al., 2021, 2019b, 2019a; Soni and Syed, 2021) and modelling of regional climate, which can significantly be modulated by irrigation (Mishra et al., 2020; Thiery et al., 2020). Ultimately, such improved knowledge will support policy makers to make valid and timely decisions on water management (Schwartz et al., 2020).

Soil moisture-based irrigation estimates have been found to yield irrigation estimates with satisfactory accuracy (Brocca et al., 2018; Dari et al., 2020; Zaussinger et al., 2019). However, the advantages of using ET over soil moisture is: 1) ET is directly linked to plant transpiration reacting to irrigation whereas soil moisture produces an indirect estimate, especially since many remote sensing systems only penetrate the topsoil (few cm), 2) the spatial resolution is higher of readily available ET datasets (e.g. derived from optical and thermal MODIS data). The disadvantage of using ET to estimate irrigation is that
the magnitude of the rainfed component of the products can vary substantially, which can in theory led to diverging irrigation estimates when comparing across ET products. Similar to Koch et al. (2000), we used a hydrological model calibrated for rainfed conditions to simulate a rainfed baseline and thus accommodate for the differences between ET products.

Less attention has been given to quantify the uncertainty of ET-based irrigation estimates. Uncertainties can be expressed twofold, i.e. accuracy and precision. Accuracy captures how close the estimates are to observations, whereas the
precision investigates how close or dispersed estimates are to each other. Accuracy of irrigation estimates can only be assessed by observations, which are commonly absent at larger scale. In this study we focus on precision, which can be addressed by means of an ensemble approach, utilizing multiple models, i.e. with different hydro-meteorological datasets.

Although remote sensing-based hydro-meteorological data have the advantage of high spatial coverage, inherent uncertainty in ET and precipitation products may arise from a variety of potential errors (e.g. different revisit time from satellite
sensors and model approach). Evaluation of evapotranspiration products by the water balance ET and Budyko ET approach in Africa and Europe have shown that ET remote sensing products may differ substantially when comparing magnitude and/or spatial patterns (Stisen et al., 2021; Weerasinghe et al., 2020). Evaluation of precipitation products have, analogous to the ET products, shown that large differences in magnitude and spatial patterns are evident. For example, Yang and Luo (2014) evaluated the performance of three precipitation products over an arid region in China and found that corrections were
necessary as the products yielded very different magnitudes and spatial patterns. Logah et al. (2021) found that the precipitation products generally performed better during the dry period and that the products had difficulties simulating high intensity rainfall in the Black Volta Basin.

The current study area covers the Indus and Ganges basins, shared between more than a billion people in India, Pakistan, Nepal, Bangladesh, China and Afghanistan. Large government investments in India in the 1960 led the region and
mainly the state of Punjab to be the largest area heavily equipped for irrigation at global scale, by construction of the Indus



Basin and Bhakra irrigation systems, providing food security beyond its borders (Sharma et al., 2010). A rapidly growing population, combined with a decreasing investment in irrigation infrastructure have increased unsustainable groundwater use and resulted in a regional decline of groundwater level (Rodell et al., 2009). A regional survey indicated that irrigation from groundwater was more widespread than first assumed as only 5% of surveyed villages consider their agricultural practice as
totally rainfed (Shah et al., 2006).

This study applies, for the first time, an ensemble approach to investigate the robustness of ET-based estimates of irrigation at regional scale for a global hotspot of irrigation induced groundwater overexploitation. In this way, previous work (Koch et al., 2020; Romaguera et al., 2014), is expanded by using different ET and precipitation products to quantify irrigation water use and precision of an ET-based framework. The three main objectives of this paper are: 1) the selection and analysis
of suitable global ET and precipitation dataset for the irrigation quantification over Indus and Ganges basins, 2) building a hydrological model to simulate rainfed ET at 5 km spatial resolution via a state-of-the-art calibration tool and 3) evaluate the precision and influence of ET and precipitation uncertainties in the estimation of irrigation.

## 2 Study area

Indus and Ganges basins extend over an area of 2.2 million km$^2$ (Figure 1). The region can be subdivided into four geographical
regions: 1) The Himalayan mountains along the northern boundary, 2) The Indo-Gangetic outwash plain, 3) The Thar desert separating the two basins; and 4) The peninsular plateau south of the Indo-Gangetic plain, characterized by highlands, valleys and rounded hills. Climate is monsoon dominated and varies from a tropical humid zone in the eastern Ganges basin and along the mountain range, to an arid climate in lower Indus basin (see Figure 1). Most precipitation occurs from July to September during the monsoon season and varies on average between 200-1200 mm/year (2000-2019) across the basins.
Agriculture accounts for 70% of land cover in the basins. Summer rice and winter wheat rotation is the most common cropping system in the Indo-Gangetic plain, mixed with cotton and sugarcane outside the plain (Cai et al., 2010). Summer rice water requirements are overall met by precipitation during the wet period (May – November), except in lower Indus basin with precipitation rates less than 50 mm/month, where extensive irrigation takes place also during the monsoon months. However, winter wheat heavily depends on irrigation in the entire region as the average precipitation rate is less than 25 mm/month
during the dry period (December-April).

## 3 Method and data

### 3.1 Hydrological model

This study applies the grid-based mesoscale Hydrological Model (mHM, Kumar et al., 2013; Samaniego et al., 2010; Thober et al., 2019) version 5.11.0 (Samaniego et al., 2021). mHM uses a multiscale parameter regionalization technique that links
spatial distributions of model parameters at an intermediate scale, representing hydrological processes, to finer scale variability





in soil texture, topography and vegetation via nonlinear transfer functions. The transfer functions have a limited number of global parameters that enable an efficient calibration (Samaniego et al., 2021, 2017). The hydrological models set up for this study used 10 km gridded metrological forcing and 1 km morphological data and were calibrated and executed at 5 km spatial resolution to simulate natural ET baselines, i.e. representing a purely rainfed hydrological system without the presence of
irrigation.

For our model setup, ET is calculated as a fraction of potential ET by applying the Fedde's reduction factor (Feddes et al., 1976). mHM offers an option for dynamic downscaling of potential ET from metrological- to model-resolution by incorporating vegetations dynamics from a monthly leaf area index (LAI) climatology (Demirel et al., 2018). In order to setup mHM to simulate rainfed ET, the LAI climatology had to be corrected by removing the imprint irrigation has on the vegetation
dynamics. Therefore, we created a rainfed cropland mask (described in section 3.4) to calculate mean rainfed cropland LAI climatologies for each climate zone during the irrigation cropping season. The rainfed LAI climatologies were used to replace the irrigated LAI climatologies to simulate rainfed ET baselines. Irrigated LAI climatologies in the arid climate zone were replaced with the rainfed LAI climatology from the semi-arid zone as no rainfed cropland were identified in this area.

In this study, different precipitation products were used as forcing (described in section 3.3), daily average air
temperature was acquired from ERA-Land, and potential ET was calculated by using FAO-56 Penman-Monteith equation with ERA5-Land variables (Muñoz Sabater, 2019). The DEM was obtained from NASA's Shuttle Radar Topography Mission data (Jarvis et al., 2016). Soil texture information was processed for six horizons from the SoilGrid$^{TM}$ database (ISRIC, 2020) and resampled to 1 km using the mean function. LAI and land cover data were collected from MODIS MCD15A2H.v006 and MCD12Q1.v006, respectively.

**3.2 Calibration strategy**

The calibration framework is designed to obtain hydrological models that simulate baselines of rainfed ET. The hydrological models used in this study were calibrated using the Pareto Archived Dynamically Dimensioned Search (PADDS) algorithm (Asadzadeh and Tolson, 2009) implemented in the Optimization Software Toolkit – OSTRICH (Matott, 2017). The calibration was performed with 600 iterations and a perturbation size of 0.2. We calibrated 12 parameters that were identified based on a
prior sensitivity analysis perturbing one parameter at a time and recording the change in objective function. The OSTRICH algorithm provides the modeler with a pareto front of dominant solutions, which enable the modeler to select the solutions that marks the most acceptable tradeoff between multiple objective functions.

OSTRICH was used to minimize two objective functions that address the magnitude and the spatial pattern of ET over rainfed cropland and naturally vegetated areas. First, the mean absolute error (MAE) is used to target the magnitude of
ET over rainfed cropland.

$$MAE = \frac{\sum_{i=1}^{n}|x_i - y_i|}{n},$$    (1)



Where $x_i$ and $y_i$ represent observed and simulated ET at cell $i$, and $n$ is the number of cells. MAE has an optimal value of 0 and varies from 0 to positive infinity. Second, optimization of the spatial ET pattern was targeted by applying the spatial efficiency (SPAEF) metric on ET in rainfed cropland and naturally vegetated areas for the dry and wet period. SPAEF is a multi-component bias-insensitive spatial pattern metric, evaluating the models ability to simulate the observed correlation, variance and histogram. (Demirel et al., 2018; Koch et al., 2018)

$$SPAEF = 1 - \sqrt{(\alpha - 1)^2 + (\beta - 1)^2 + (\gamma - 1)^2} \, , \tag{2}$$

$$\alpha = \rho(x, y) \quad \text{and} \quad \beta = \left(\frac{\sigma_x}{\mu_x}\right) / \left(\frac{\sigma_y}{\mu_y}\right) \quad \text{and} \quad \gamma = \frac{\sum_{j=1}^{n} \min(K_j, L_j)}{\sum_{j=1}^{n} K_j}$$

Where $x$ and $y$ denote observed and simulated data. $\alpha$ is Person's correlation coefficient, $\beta$ is the spatial variability, calculated as a fraction of observed and simulated coefficient of variation and $\gamma$ is the agreement between the observed ($K$) and simulated ($L$) histograms with $n$ bins. SPAEF has an optimal value of 1 and varies from 1 to negative infinity In order for OSTRICH to minimize the SPAEF objective function, we calculated the sum of squared residuals for dry and wet periods.

To select the best parametrizations after having obtained the full pareto front from OSTRICH, we normalized each dominant solution in the pareto front by the best performance for MAE and SPAEF. The solution with the lowest sum was then selected for each pareto front as the best parametrization. Because the ranges in MAE are larger than ranges for SPAEF, the MAE dimension was truncated by minimum dominating MAE plus 1 mm/month.

### 3.3 Evapotranspiration and precipitation data

We compared seasonal and annual differences and normalized spatial patterns among ten ET products and eight precipitation products to identify the most suitable datasets for our modeling study. The precipitation data were used as forcing to the developed hydrological models and the ET data were used twofold, first as calibration target over the rainfed areas and second as reference in the subsequent irrigation quantification. An initial comparison revealed large differences across the ET products, which to a large degree were coinciding with climate zones. In contrast, differences were small among precipitation products. The final selection of ET product was based on two criteria: 1) capturing dry-period irrigation resulting in high ET during the months (December-April) and 2) realistic annual estimates (no references several orders of magnitude higher or lower than annual precipitation) with reasonable inter annual variations (no sudden changes in mean annual ET, can happen if the reference is a composite of other references). As relative differences among precipitation products were small, the sole criterion for selection was the spatial resolution, i.e. high-resolution products were favored (<0.25°). After the initial comparison of datasets, three ET and five precipitation products (Table 1) were selected for building 15 hydrological models, each calibrated based on a unique combination of the selected products.

The five selected precipitation inputs are CHIRPS , ERA5-Land, MSWEP, PERSIANN-CDR and TRMM (Table 1): CHIRPS uses reanalysis and satellite infrared data to estimate precipitation and gauge observations for correction (Funk et al.,





2015). ERA5-Land is a high spatial resolution land component of the global ERA5 climate reanalyses system; a product driven by a large amount of satellite and gauge data (Muñoz Sabater, 2019). MSWEP is a synthesis of different precipitation products
that are merged using gauge observations (Beck et al., 2019). PERSIANN-CDR is a machine learning product that uses satellite infrared data and gauge observations for bias correction (Ashouri et al., 2015). TRMM use infrared and microwave satellite data to estimate precipitation and gauge observations for subsequent correction (Huffman et al., 2007). Precipitation products are very similar when comparing seasonal and annual variations and showed one distinct peak during the summer monsoon (Figure 2C and D). However, relative differences of up to 40% were found between the annual precipitation rates in the arid
climate zone in the lower Indus Basin.

    The three selected ET products are FLUXCOM, NTSG and PML (Table 1): FLUXCOM is a machine learning product that combines energy balance observations at flux towers with satellite data (Jung et al., 2019). NTSG is a satellite and reanalysis driven product that combines the Penman-Monteith and Priestley-Taylor models (Zhang et al., 2010). PML is a satellite and reanalysis driven product that is based on the Penman-Monteith and Leuning models (Zhang et al., 2019). All
three products have in common that they utilize to a large degree thermal and optical data from MODIS. The ET products were rather different with respect to their seasonal and annual variations, but were overall characterized by two distinct peaks, the first in March and the second between July-September (Figure 2A and B). The seasonal pattern is dominated by the summer monsoon and influenced by extensive irrigation during the dry period (December-April). The ET products were more similar during the dry period compared to the wet period and relative differences were observed in annual ET across the basins in the
humid (20%) and arid (50%) climate zones. Beside seasonal patterns, annual estimates suggest that ET and precipitation have increased since 2001 (Figure 2B and 2D) which agrees with other studies (Jin and Wang, 2017; Katzenberger et al., 2021).

    The selected products (Table 1) include different temporal and spatial resolutions and all have been pre-processed to the same spatiotemporal dimensions before modelling. ET and precipitation products have been aggregated by summation to monthly and daily scales, respectively. Further, all ET products have been up- or downscaled to 5 km, and precipitation data
were resampled to 10 km spatial resolution by bilinear interpolation.

**Table 1: Characteristics of selected ET and precipitation products.**

Abbreviations: ECMWF Reanalysis 5[th] Generation – enhanced resolution (**ERA5-Land**), Numerical Terradynamic Simulation Group (**NTSG**), Penman-Monteith-Leuning v.2 (**PML**), Climate Hazards Group InfraRed Precipitation with Stations (**CHIRPS**), Multi-Source
Weighted Ensemble Precipitation v.2 (**MSWEP**), Precipitation Estimation from Remotely Sensed Information using Artificial Neural Networks - climate data record (**PERSIANN-CDR**) and Tropical Rainfall Measuring Mission Multi-satellite Precipitation Analysis v.7 (**TRMM**).

| | Dataset | Spatial resolution | Spatial coverage | Temporal resolution | Temporal coverage | Reference |
|---|---|---|---|---|---|---|
| Evapotrans-piration | ERA5-Land[1] | 0.1° | global | daily | 1981 - now | (Muñoz Sabater, 2019) |
| | FLUXCOM | 0.083° | global | 8-day | 2001 - 2015 | (Jung et al., 2019) |
| | NTSG | 0.083° | global | daily | 1982 - 2013 | (Zhang et al., 2010) |
| | PML V2 | 0.005° | global | 8-day | 2002 - 2019 | (Zhang et al., 2019) |





| | | | | | | |
|---|---|---|---|---|---|---|
| Precipitation | CHIRPS | 0.05° | 50°N-50°S | daily | 1981 - now | (Funk et al., 2015) |
| | ERA5-Land | 0.1° | global | daily | 1981 - now | (Muñoz Sabater, 2019) |
| | MSWEP | 0.1° | global | 3-hourly | 1979 - 2017 | (Beck et al., 2019) |
| | PERSIANN-CDR | 0.25° | 60°N-60°S | 3-hourly | 1983 - now | (Ashouri et al., 2015) |
| | TRMM | 0.25° | 50°N-50°S | 3-hourly | 1998 - now | (Huffman et al., 2007) |

[1] ERA5-Land ET is only used for validation of concept.

### 3.4 Rainfed map

To calibrate the hydrological model against rainfed conditions (cropland that are not under irrigation), we created a map differentiating rainfed and irrigated cropland. The classification of cropland into rainfed and irrigated was based on MODIS land cover and NDVI products (MODIS MCD12Q1.v006 and MOD13Q1.v006). We found inspiration from Dari et al. (2021), who used results from a temporal stability analysis of satellite and modelled soil moisture, in an unsupervised K-means analysis to detect and map irrigated areas. In our adopted approach, we used mean dry period NDVI climatologies (i.e. five months,

December-April) in a temporal stability analysis. More precisely, we used the standard deviation of the spatial anomalies  and the temporal anomalies in a 2-dimensional unsupervised K-means classification to identify three clusters representing rainfed cropland, irrigated cropland and mixed – more information about the temporal analysis components can be found in (Dari et al., 2021). The assumption is that NDVI of rainfed cropland can be characterized by a high temporal stability and a low temporal anomaly in the five selected months, and vice versa for irrigated cropland. The classification was performed at the

original MODIS resolution of 500 m and then the classification was upscaled to model resolution, i.e. 5 km. A threshold of 95% was used to identify primarily rainfed and irrigated pixels (to avoid a mixed rainfed and irrigated signal in the calibration), thus a third class was added to represent pixels that were mixed. The classification was evaluated against the FAO GIMA v5.0 dataset (Siebert et al., 2013) on global areas equipped for irrigation (Figure 1) and showed overall consistency. During the wet period, cropland classified as 'humid' according to the dryland classification by the Joint Research Center of the European

Commission (Spinoni, 2015) was assumed to be rainfed cropland (Figure 6E). The dry and wet period rainfed maps (Figure 3) were used to correct the rainfed grids in the LAI climatologies, as described in section 3.1.

### 3.5 Net irrigation estimation

Net irrigation is the amount of supplied irrigation that is lost through ET. We assume that net irrigation can be quantified as the difference between a ET reference, obtained from e.g. remote sensing, and a hydrological model acting as rainfed baseline

(Koch et al., 2020). Net irrigation is quantified on a monthly timescale and at 5 km spatial resolution for the 15 ensemble members, which are based on combinations of three ET and five precipitation products. We further assumed that by calibrating the 15 hydrological models against rainfed ET we can simulate rainfed baselines for the entire model area that match the unique combination of ET and precipitation product. Our assumption is supported by the strong parameter regionalization schemes incorporated in mHM, which link model parameters to fully distributed catchment characteristics. This will yield


physically meaningful parameter fields, which we believe are the foundation to make robust predictions of a rainfed baseline ET, also over irrigated areas. The magnitude of the ET products varies substantially (Figure 2) and we hypothesize that calibration will enable the hydrological model to accommodate this; resulting in hydrological models with different magnitudes of rainfed ET to match the differences in the reference ET products. Uncertainties can be expressed as precision and accuracy. Precision investigates the ensemble dispersion whereas accuracy is the closeness between estimates and observations. Thus,

in absence of observations, the accuracy of our net irrigation estimate cannot be quantified. Nevertheless, we believe that analyzing the precision of irrigation estimates is a valuable and novel contribution. We define net irrigation as the difference between ET as obtained from the reference products and the rainfed hydrological baseline model:

$$net\ irrigation = ET_{reference} - ET_{baseline} \hspace{3cm} (3)$$

For rainfed areas it is assumed that $ET_{reference}$ is equal to the $ET_{baseline}$, thus for irrigated areas $ET_{reference}$ is expected to

exceed the $ET_{baseline}$ resulting in positive residuals (net irrigation). Negative residuals are a sign of an overestimation of the rainfed hydrological model. If occurring, negative residuals can be related to uncertainties in the precipitation forcing, the ET product used as reference or the hydrological baseline model itself.

### 3.6 Variance decomposition analysis

The model ensemble yielded 15 different net irrigation estimates and we applied a variance decomposition analysis to

investigate the sources of uncertainties in more detail. The uncertainty contribution from the two investigated sources, namely ET reference and precipitation on net irrigation was analyzed following the approach of Déqué et al. (2007). This analysis quantifies the magnitude of net irrigation variance caused by the two uncertainty sources, thus ranking the influence of ET and precipitation. The procedure of the method is: 1) calculate the variance contribution from both uncertainty sources and contribution from interactions between sources, thus the total variance is the sum of all three variance contributions, 2)

for each uncertainty source calculate the variance term, as percentage of the total variance, by summing the individual source variance and contributions from interactions and then dividing by the total variance. The sum of the two variance terms is more than the total variance as the latter includes both the individual source variance and contributions from interactions between sources, but the magnitudes of the two variance terms indicate the individual role of each uncertainty sources on the total variance (Déqué et al., 2007). The analysis was applied on monthly net irrigation estimates for each climate zone. The

variance decomposition analysis has successfully been applied in a range of hydrological applications, for example to study the uncertainty contributions of climate model and hydrological model structure on climate change impact simulations (Karlsson et al., 2016).



## 4 Results and Discussion

### 4.1 Baseline models

The pareto fronts based on the 15 calibrations conducted (Figure 4) show the tradeoff between the two applied objective functions for rainfed ET, namely MAE addressing the magnitude of ET and SPAEF addressing the spatial pattern performance. We tested different numbers of iterations and perturbation sizes prior to the calibration and based on our findings we expect a higher number of iterations (more than 600) to only marginally improvement the tradeoff around the optimal solution, but primarily extent the tails of the pareto fronts. In general, the range in MAE of the pareto fronts are larger than for SPAEF

because we assume that model parameters can easily change the ET magnitude, but the simulated bias insensitive spatial patterns are as a starting point more realistic. This is because the simulated spatial patterns are to a large degree linked to the spatial parameter fields which again are tied to fully distributed catchment characteristics, such as soil and vegetation variability. This will limit the range of SPAEF and rule out very poor pattern performance.

Based on the pareto fronts, the tradeoff between the two applied objective functions can be studied and we selected a

single optimal parametrization for each of the 15 baseline models using the approach described in section 3.2.  The MAE of the 15 selected runs lie within a range of 13-17 mm/month and the SPAEF varies between 0.44-0.76 during the dry period and between 0.60-0.85 during the wet period. The baseline models calibrated against NTSG ET reference vary from the remaining models by having a SPAEF that ranges between 0.44-0.63 during the dry period and between 0.70-0.74 during the wet period, and thereby show the poorest spatial pattern performance. This shortcoming relates to the homogeneous pattern in satellite-

based ET reference during the pre-monsoon period in April-May, which the baseline models cannot simulate. The baseline model calibrated against ERA5-Land reference and uses ERA5-Land precipitation is plotted as a 16. pareto front in Figure 4. ERA5-Land does not directly incorporate irrigation and we have included this calibration as proof-of-concept that our calibration approach can reproduce a rainfed hydrological model. For this calibration, climate input and calibration target are obtained from the same modeling system and are therefore in good agreement.

The ensemble ET baselines vary in a range of 256-455 mm/year for Indus and 468-667 mm/year for Ganges basins, which is according to the ET references they were calibrated against. This implies that the ensemble baseline of rainfed ET is just as uncertain as the ET references, but the aim is not to simulate the actual rainfed ET but to finetune each baseline hydrological model to their satellite-based ET reference and hereby enable a subtraction of rainfed ET from irrigated areas. A large range in ensemble baseline thus indicates that the calibration has served its purpose. Kushwaha et al. (2021) used an

ensemble of hydrological models and applied the Budyko approach to estimate ET across the Indian sub-continental river basins and found ET in Indus and Ganges in the range of 246-369 mm/year and 511-622 mm/year, respectively.

The spatial patterns of the ET baselines are characterized by high ET along the Himalayan mountains and a regional East-West gradient matching the climatic zones (Figure 5A and C). This emphasizes that the baselines simulate rainfed ET according to precipitation patterns (Figure 5C and D). It becomes obvious that the ERA5-Land reference (Figure 5B) does not

consider the effect of irrigation on ET in Indus and Ganges and we found that only minor parts of the cropland is classified as



irrigated in the ERA5 reanalysis model (ECMWF, 2018). Since irrigation does not affect ERA5-Land, the spatial patterns of the ERA5-Land baseline (simulated by mHM) and ERA5-Land ET reference (Figure 5A and B) are expected to match also for irrigated areas. We calculated SPAEF between ERA5-Land baseline and reference ET for rainfed and irrigated cropland and found SPAEF for rainfed cropland to be 0.79 and irrigated to be 0.88, which means that baseline and reference ET matches

well in both rainfed and irrigated areas. We found that the ERA5-Land baseline was able to reproduce the natural precipitation induced ET patterns in the irrigated areas but have minor elevated ET in the desert due to model uncertainty.

This underpins the validity of the method, i.e. that a hydrological model can be calibrated to reproduce rainfed ET originating from an alternative reference. By comparing FLUXCOM baseline and reference (Figure 5C and D), the ET baseline magnitude is similar to the ET reference for rainfed areas and the spatial pattern resembles precipitation patterns. Thus, the

hydrological model can simulate a realistic rainfed ET baseline.

## 4.2 Net irrigation ensemble estimates and precision

The analysis is based on an ensemble of 15 independent net irrigation estimates (from now on referred to as ensemble estimates). The main finding of the analysis is that the standard deviation of the ensemble estimates is low in most of the study area (Figure 6B). Although the ensemble baselines, i.e. the simulated rainfed ET of the 15 models, differ by about 91.5

mm/year, the net irrigation precision is 42.5 mm/year for the entire region.

The ensemble estimates of the dry period (Figure 6A) show high net irrigation across the Indo-Gangetic plain. Net irrigation is largest in the northern Punjab region as expected (Sharma et al., 2010) and a decrease from West to East following the transition from arid to humid climatic zones (Figure 6E) can be observed. Dry period ensemble estimate precision is evenly distributed across all four climate zones (Figure 6B), illustrating the importance of calibration to obtain comparable net

irrigation magnitudes from references with different ET magnitudes. The wet period ensemble estimate (Figure 6C) shows high net irrigation in the arid zone, which we did not expect. The precision is highly correlated in space during the wet period expressed by a cluster of low precision, i.e. high standard deviation, in the arid zone (Figure 6D). Based on further analysis, we relate this effect to the apparent overestimation of FLUXCOM and PML ET references in the lower Indus basin. Both products assign a high magnitude, uniform ET across the entire arid zone.

The temporal variation of the ensemble estimates and their precision (Figure 7) show that net irrigation estimates peak during February-March in the entire region and that precision is well defined at a monthly scale, except in the arid zone during the wet period (Figure 7A). The mean ensemble estimate and precision in Indus is estimated to 245.9±76.0 mm/year (78.4±24.5 km$^3$/year) and mean ensemble estimate and precision in Ganges is estimated to 115.1±25.7 mm/year (75.7±16.9 km$^3$/year) (Table 2). This underlines the higher intensity of irrigation in the Indus basin as the total irrigation water use is about

the same as the Ganges basin despite the substantially smaller cropland area (Indus 796.8 million ha, Ganges 1643.4 million ha). Aggregated seasonal ensemble estimates indicate that net irrigation in the Indus basin is evenly split between the dry and wet period (54 and 46% respectively), whereas 75% of net irrigation in the Ganges basin occurs during the dry period. The mean ensemble estimate and precision aggregated for both, Indus and Ganges basins is estimated to 157.7±42.5 mm/year





(153.9±41.5 km³/year), thus a precision of 27% of the total irrigated water use. By comparing basin and regional ensemble
estimates, the regional estimate is influenced by the lower precision in the Indus basin during the wet period. We therefore
want to highlight a precision of 16% (25.0 mm/season) in both basins during the dry period (Table 2).

**Table 2. Overview of ensemble net irrigation estimates and precision for Indus and Ganges basins separately and aggregated as a region. The wet period net irrigation and precision is calculated according to dry period irrigated cropland.**

| Unit | Total irrigation (mm/year) | Total irrigation (km³/year) | Wet period irrigation (mm) | Dry period irrigation (mm) | Yearly precision (mm/year) | Yearly precision (km³/year) | Wet period precision (mm) | Dry period precision (mm) |
|---|---|---|---|---|---|---|---|---|
| Indus | 245.9 | 78.4 | 113.5 | 132.5 | 76.0 | 24.5 | 60.4 | 27.9 |
| Ganges | 115.1 | 75.7 | 29.3 | 85.8 | 25.7 | 16.9 | 11.3 | 23.7 |
| Indus & Ganges | 157.7 | 153.9 | 56.7 | 101.0 | 42.5 | 41.5 | 27.2 | 25.0 |


The mean monthly standard deviation was found to depend on the climatic zones and decreased from 8 to 4 mm/month
during the dry period and from 12 to 4 mm/month during the wet period as the aridity index increases, i.e. going from arid to
humid climate. This overall increase in precision across the four climate zones (Figure 7A to D) coincide with a decrease in
ET reference uncertainty. Estimating ET can be very difficult under extreme climatic conditions such as arid zones and is
strongly depended on the modelling approach (Jung et al., 2019; Zhang et al., 2019). This is also evident for our initial analysis
of ten different reference models. Comparing seasonal coefficients of variation show that the standard deviation is 35% of the
mean net irrigation during the wet period and 22% during the dry period, which is consistent in both basins. Lower precision
during the wet period has been reported for irrigation quantifications using alternative soil moisture-based approaches
(Jalilvand et al., 2019; Zohaib and Choi, 2020) and result from less irrigation being used to supplement precipitation during
the wet period, whereas irrigation largely replace precipitation during the dry period. Therefore, it can be difficult to isolate
the net irrigation signal from ET affected primarily by precipitation during the wet period.

The uncertainty of the rainfed ensemble baselines are evaluated based on ET residuals over rainfed cropland that have
a mean error of 32.5 mm/year, which correspond to a 5.2% error. This low bias implies that the baseline models were able to
reliable simulate rainfed ET that matches the ET references and can be understood as a measure of accuracy under the
assumption that the simulation bias over rainfed cropland can be transferred to irrigated cropland. For irrigation quantification
of the North China Plain, Koch et al. (2020) found that the accuracy was highest during the monsoon season due to energy
limiting conditions. We found the accuracy to be equally high in both wet and dry periods. We assume, that this is due to the
skewed weight on wet period rainfed cropland during the calibration, as this area is much larger than dry period rainfed
cropland (Figure 3). The precision of the ensemble estimates (42.5 mm/year) can be attributed to ET and precipitation
uncertainties and the accuracy (32.5 mm/year) can be attributed to uncertainties originating from the hydrological model, ET



references as well as precipitation uncertainties. This implies that the precision and accuracy are not independent in our case and that that the total variance is not simply the sum of the two.

Comparison of irrigation estimates can be challenging as notions might cover different aspects like irrigation water withdrawal, irrigation water requirement or net irrigation as the ET loss to the atmosphere. Water statistics from the
AQUASTAT database estimated a yearly irrigation water requirement in Pakistan (126 km$^3$/year) and India (370 km$^3$/year). The estimates are based on climatic conditions and crop physiological processes and encompasses all water to meet crop water requirements, water for flooding of paddy fields, water for land preparation etc. (Frenken and Gillet, 2012). Based on the assumption that yearly irrigation water requirement estimated by AQUASTAT is true, our net irrigation estimates suggest that about 31% of the total irrigation water requirement for the entire Indian sub-continent is lost through ET in the Indus and
Ganges basins.

We found the difference, due to irrigation in cropland, between baseline and reference ET to be 58% and 16% in Indus and Ganges, respectively. However, a 58% increase might be an overestimation that arise from the FLUXCOM and PML references. If only considering NTSG baseline and reference ET the change in Indus is found to be 42%, which seems to be more appropriate. Shah et al., 2019b used a soil moisture deficit approach and estimated a percent change in ET between
a natural and irrigated scenario modeled with the Variable Infiltration Capacity model. They found annual ET from 1951-2012 to increase by 47% and 12% in Indus and Ganges because of irrigation activities, respectively. The mismatch to our reported figures could result from their irrigation timing being off and hereby allowing irrigation to occur in between harvest and sowing when the fields are fallow, but overall a good match of results. Shah et al. (2019a) incorporated reservoirs and irrigation water demand into the model framework from Shah et al. (2019b) and found ET to increase about 16.1% and 15.7% in Indus and
Ganges, respectively. Our results compare well with this estimate for Ganges. In both studies (Shah et al., 2019b, 2019a), the natural model seems to be the calibrated against data that potentially could be influenced by irrigation like irrigation water demand – only (Shah et al., 2019a) – and streamflow, which could underestimate ET in a managed scenario.

### 4.3 Influence on ensemble precision

The main finding of our variance decomposition analysis is a strong control of ensemble estimate variance by ET. ET account
for 81% of ensemble estimate precision across the basins and the influence of precipitation is observed to increase in more humid climate zones (Figure 7, blue and yellow bars). However, the contribution of precipitation becomes more prominent in the monsoon season from July-September and around March (Figure 7) and thus tends to follow the precipitation climatology (Figure 2C).

The ET reference and any related uncertainties affect the baseline ET estimates through the calibration and the net
irrigation estimation as the baseline ET is subtracted from the reference ET. On the other hand, precipitation uncertainty only affects the baseline ET models. Thereby reference ET directly affects the net irrigation estimates whereas precipitation uncertainty acts indirectly as it is propagated through the hydrological model to impact the baseline ET. Furthermore,





precipitation uncertainty between irrigated and rainfed cropland is likely similar, whereas uncertainty between irrigated and rainfed ET may vary in the reference ET products.

Thus, it is difficult to conclude whether the influence of precipitation increases because of the uncertainty or it increases because the ET uncertainty decreases. The fact that the influence of precipitation tend to follow the seasonal variation in precipitation emphasizes that ET residuals are more difficult to extract during high precipitation (Koch et al., 2020). In the arid zone, the influence of ET is higher during the wet period, which is due to the high ET uncertainty and potential errors in FLUXCOM and PML. The ET uncertainty seems to overrule the high precipitation uncertainty in the arid zone even though

ERA5-Land and MSWEP precipitation inputs are about 40% lower than the other precipitation inputs.

**6 Conclusion**

This study focusses on an ET-based approach to estimate irrigation water use for the Indus and Ganges basins, a global hotspot of unsustainable irrigation practice. We investigated the influence of different ET reference models and precipitation inputs on precision of irrigation estimates by analyzing an ensemble of 15 net irrigation estimates. We showed that isolating the

irrigation component through ET residuals of rainfed ET baselines and reference ET models yield high precision estimates of net irrigation.

- We estimated net irrigation of the Indus and Ganges basins to 157.7±42.5 mm/year (153.9±41.5 km$^3$/year), of which about half of the irrigation takes place in the Indus basin despite accounting for only 35% of the irrigated cropland

areas.
- We found that even though ET varied by 91.5 mm/year between reference ET products, precision of net irrigation was just 25.0 mm/season during the dry period.
- We found that net irrigation precision increased as reference ET uncertainty decreased, which related to the climatic conditions of the area.

- We found that ET accounted for 81% of net irrigation variance and that the influence of precipitation uncertainty was highest during the monsoon season from July-September.

We emphasize the strength of model calibration to compensate for ET biases to create robust net irrigation estimates. As large differences in seasonal and annual rainfed ET may be evident between reference models, the magnitude of ET

variation induced by irrigation within each ET reference yield net irrigation estimates of comparable magnitudes. Therefore, it is essential to calibrate and finetune each baseline model to a reference rainfed baseline to extract net irrigation.





**Author contribution**

SJK, JK, SS and RF designed the study and SJK carried it out in close consultation with JK. SJK prepared the manuscript and figures in close consultation with JK. All authors discussed results throughout the study period and provided critical feedback

to the manuscript drafts and approved the final version of the manuscript.

**Competing interests**

The authors declare that they have no conflict of interest.

**Acknowledgements**

This study was funded by the Independent Research Fund Denmark, project number: 0164-00003B.

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







Figure 1: Map of climate zones, by the Joint Research Center of the European Commission (Spinoni, 2015) and area equipped for irrigation as percentage of area(Siebert et al., 2013). Overview figure, in top-right panel, shows the location of the Indus and Ganges basins and rivers.





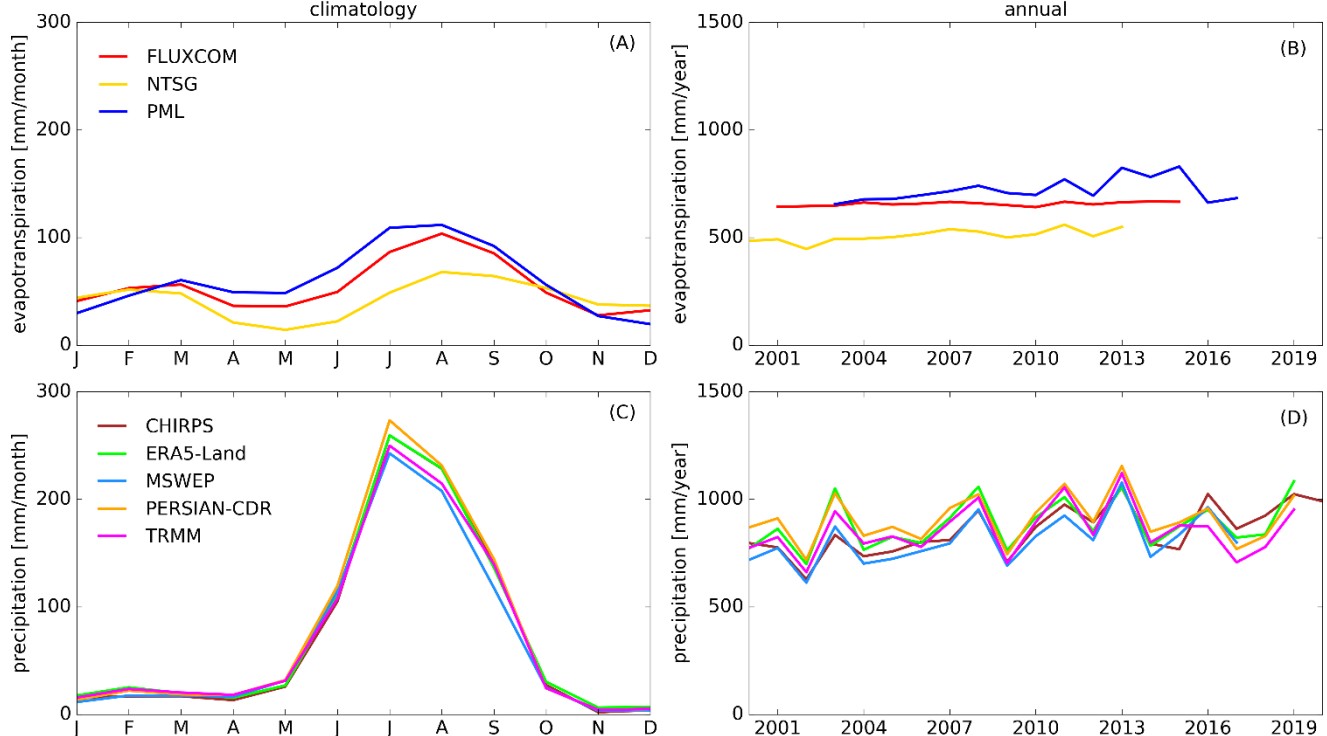

**Figure 2 Regional climatologies and annual estimates of three evapotranspiration references (A) and (B), and five precipitation inputs (C) and (D) for the entire study area. Climatologies are based on avalible data from 2000-2020.**

695





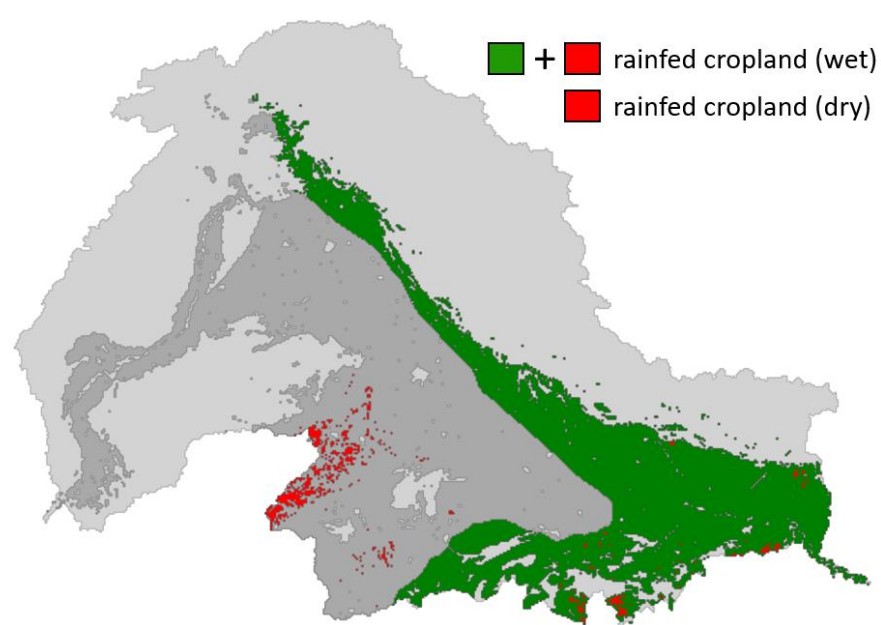

**Figure 3: Map showing the classification of rainfed cropland applied in the evapotranspiration calibration during dry (red) and wet (red and green) period. Light gray signature deliniates the Indus and Ganges basins whereas the drak gray signature shows irrigated cropland in both dry and wet period. Green indicates cropland that is only irrigated in the dry period.**

700


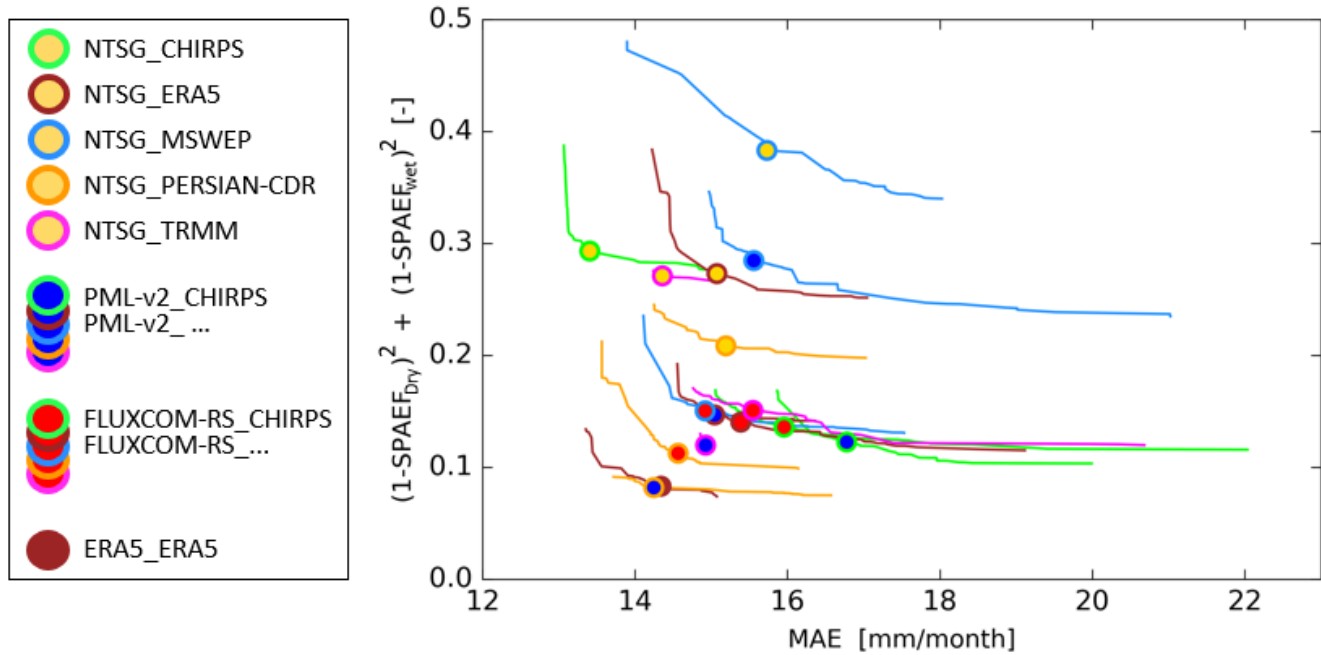

**Figure 4: Calibration results for the 15 baseline models regarding the two defined objective functions: MAE and SPAEF. The lines represent the pareto fronts, containing the dominant solutions, and the points the selected parametrizations with the optimal trade-off between objective functions. Point colours represent the three reference models and line colours represent the five precipitation inputs. Colour scheme is consistent with the legends in Figure 2.**

**Figure 5: Average modelled baseline (left panel) and reference evapotranspiration (right panel) for Febuary, 2004. Both baseline models use ERA5-Land precipitation input.**





**Figure 6: Mean ensemble net irrigation estimates (left panel) and ensemble standard deviation (right panel) for dry period (upper panel) and wet period (lower panel). E: Dryland classification by the Joint Research Center of the European Commission (Spinoni, 2015), red: arid, orange: semi-arid, yellow: dry, green: humid.**





**Figure 7: Temporal ensemble net irrigation estimates and precision for each climate zone, A: arid, B: semi-arid, C: dry and D: humid. The solid line indicates the mean monhtly net irrigation whereas the shaded envelope the precision as +/- 1 standard deviation. Lower bar chartes illustrate results from the variance decomnposition analysis and show to what degree evapotranspiration (ET, yellow) and precipitation uncertainty (P, blue) explain the ensemble variance.**