# Peer review of "The precision of satellite-based net irrigation quantification in the Indus and Ganges basins"

_Hydrology and Earth System Sciences, 2022_

## Referee Comment (RC2)

**General comment:**

Recommendation: Moderate revision

The net irrigation over two severely water-stressed basins (Indus and Ganges) is estimated by subtracting the satellite-based actual evapotranspiration (ET) from a baseline rainfed ET estimated through hydrological modeling. This study is a follow on to a previous study by Koch et al., 2020, with one significant enhancement of using an ensemble approach in which multiple precipitations and RS-based ET data are used to create an ensemble model simulation to estimate net irrigation and its uncertainty. The results are nicely presented and the manuscript is well-written. However, I believe the authors should distinguish between consumed irrigation by the crops (what is estimated in this study) and net irrigation water use which can be significantly higher than consumptive water use (based on irrigation efficiency). I am also concerned about replacing the observed LAI with the rainfed LAI climatology to calculate the rainfed component of ET. This can potentially lead to false baseline ET estimation by removing the irrigated crop characteristics. Details on these main concerns along with some other moderate and minor comments are provided below. Addressing all these comments I suggest the acceptance of the paper.

**Specific comment:**

**Major:**

1.  L44: There is an important distinction between the irrigation water consumed by crops and the net irrigation. ET is a measure of consumptive water use which is consumed irrigation water over the irrigated area. In many cases of flood or surface irrigation, a substantial portion of irrigation is lost to drainage (not consumed by the crops). This is especially important in your case studies where the irrigation efficiency is reported to be less than 52% on average (Simon et al., 2020). Please clearly mention in the manuscript that what is estimated here is consumed irrigation and not net irrigation.

    *Simons, G. W. H., et al. "A novel method to quantify consumed fractions and non-consumptive use of irrigation water: Application to the Indus Basin Irrigation System of Pakistan." Agricultural Water Management 236 (2020): 106174.*

2.  **The problem with replacing LAI over irrigated agriculture with climatology LAI over rainfed areas:**

    The only place where the crop or land cover type is incorporated in the actual ET estimation in the hydrological model is in the downscaling of ET potential using the LAI data. Here I quote from a reference study (Demirel, et al., 2018) that is cited here for this part of the methodology: "*The DSF (vegetation dynamic coefficient) is parametrized using spatiotemporal LAI component accounting for the effect of characteristics that separate the actual vegetation from a reference grass. These characteristics include*

*specific landcover, albedo and aerodynamic resistance ...,"* here, you are replacing the main component of crop and landcover characteristics over the irrigated area with a rainfed climatology LAI which has different characteristics (land cover, crop type, albedo, etc.). This can lead to a false baseline ET estimate and consecutively net irrigation and can be a major source of error that needs proper attention and discussion in the manuscript. Please comment on the possible impact of this replacement on the final net irrigation estimation.

3. L233: part of uncertainty can be attributed to the simplified model physics and the heterogeneity of land cover which is not mentioned in the manuscript. Please discuss these other sources of uncertainty in the manuscript as well.

**Moderate:**

1. 1.L34: there is also a more recent study on irrigation mapping using SMAP-Sentinel1 1kmsoil moisture data that can be cited here:

    *E. Jalilvand, R. Abolafia-Rosenzweig, M. Tajrishy and N. N. Das, "Evaluation of SMAP/Sentinel 1 High-Resolution Soil Moisture Data to Detect Irrigation Over Agricultural Domain," in IEEE Journal of Selected Topics in Applied Earth Observations and Remote Sensing, vol. 14, pp. 10733-10747, 2021, doi: 10.1109/JSTARS.2021.3119228.*

2. L44: some other disadvantages of using ET that are not mentioned here:
    a. Limitation of ET estimation in cloudy weather situations
    b. The ET is an estimation of consumptive water use not irrigation

3. L 62: There are many other studies on the satellite-based ET and consumptive water use estimation over the Indus and Ganges basins which can be referred to in the introduction or the discussion section of the paper. Studies such as:

    *Karimi, P., Bastiaanssen, W. G. M., Molden, D., and Cheema, M. J. M.: Basin-wide water accounting based on remote sensing data: an application for the Indus Basin, Hydrol. Earth Syst. Sci., 17, 2473–2486, https://doi.org/10.5194/hess-17-2473-2013, 2013.*

    *Simons, G. W. H., et al. "A novel method to quantify consumed fractions and non-consumptive use of irrigation water: Application to the Indus Basin Irrigation System of Pakistan." Agricultural Water Management 236 (2020): 106174.*

    *Peña-Arancibia, Jorge L., Joel P. Stewart, and John M. Kirby. "Water balance trends in irrigated canal commands and its implications for sustainable water management in Pakistan: Evidence from 1981 to 2012." Agricultural Water Management 245 (2021): 106648.*

4. L113-114: a more recent Modis product version (v 061) was introduced at least a year ago (late 2020) and the research community is advised to use this product due to changes and improvements in the calibration approach. It is expected that the most recent product is used in a study that is going to be published in late 2022. It would be interesting if a test analysis were conducted using the v061 data and the differences were reported in the supplementary material.

5. L127: is there any time dimension in the optimization conducted in this study or the optimization is only done in the space domain and on one image (Snapshot)? Can you comment on how different it would be if the optimization were conducted for each pixel separately and in time and why not time series based objective function is used in your optimization?

6. L207: net irrigation is a misleading phrase as explained in the major comment (1).

7. L290: please explain why the net irrigation precision is higher than the ensemble baseline rainfed ET.

8. L378: I assume here the Author meant RS-based actual ET by the reference ET which is again misleading as the reference ET has a different meaning in the evapotranspiration community. I suggest using different terminology.

**Minor:**

1. L19: 25 mm/season is the average of two basins? Please explicitly mention
2. L19: I think an "of" is missing after "the robustness"
3. L46: Koch et al., 2020 …
4. L261: 16th …
5. L265-266: this sentence is not clear to me please rephrase

---

## Author Response (AR1)

[Reviewer comments in normal font; *Author replies in itialic*]

**REVIEWER 1**

The manuscript provides a novel and useful evaluation of a technique to estimate net irrigation over a global irrigation hotspot (the Indus and Ganges basins). Overall, the results are impressive, with quantification of net irrigation from an ensemble of unique realization having strong agreement in most cases. I recommend publishing the manuscript after addressing the below comments and expect this will be a widely used study by the remote sensing, hydrology, and agricultural science communities.

*Reply: We thank the reviewer for the overall positive and constructive feedback that will help us to further improve our work. Below, we outline how we consider responding to the issues pointed out by the reviewer in the revision and what changes were implemented.*

**General comments**

Is the transferability of calibrated parameters evaluated in space and time (i.e., to non-calibrated areas and times?). It seems the accuracy of the methodology relies on these calibrated parameters being transferable from non-irrigated places or times to irrigated areas. It would be valuable to estimate irrigation from non-irrigated areas following the same method used to estimate irrigation over the irrigated areas to provide an estimation of the method's potential systematic biases. Similarly, it would be valuable to show an ET bias time series for the calibrated model over non-irrigated times and irrigated times, respectively (which could be a supplemental figure). It would be helpful to include more detail about the calibration procedure as well. Namely, for irrigated pixels, is the model calibrated during non-irrigated periods or are calibrations from rainfed areas transferred to irrigated areas? In either case, there needs to be an evaluation the parameter transferability.

*Reply: Thanks for raising this point. We believe that the manuscript will benefit from a better presentation of the calibration results. In the initial submission, the model is only validated in time (calibration period: 2003-2007, validation period: 2008-2012). To validate the parameter transfer in space we plan to provide model performance for a parameter transfer from rainfed to irrigated areas for a model calibrated against ERA5-Land. ERA5-land does not include irrigation and could therefore be used to validate the spatial transferability of rainfed parameters.*
*Plan for revision: Include a validation of parameter transferability in space and time and report cal/val performance of the applied metrics (SPAEF and MAE). Include supplemental figures of rainfed ET bias timeseries and maps.*
*Changes: We included table S1 with cal/val statistics for rainfed areas. We included information on calibration and validation strategy (also changed section name) (lines:148-151). We added a couple of lines about validation of spatial parameter transfer by using ERA5-Land as baseline and added a figure to supplementary material with ET bias timeseries and maps. (lines: 278-286)*

Please include how data and code can be accessed.

**Reply:** *This is a relevant point. Thank you for bringing this up. The ensemble estimates of the quantified net irrigation, and uncertainty will be made publicly available via an online repository. In addition, code and model setup will be shared upon personal request.*
**Plan for revision:** *The data acknowledgment section will be extended accordingly and the online data repository will be created.*
**Changes:** *Section have been added on data availability. (lines: 441-443)*
* * *
Please provide a thorough grammatical edit of the manuscript before re-submission.

**Reply:** *Thanks for reading or manuscript carefully. We will try to correct grammatical mistakes to the best of our abilities.*
* * *
Line 12: "an novel" should be "a novel".
Line 45: "led" should be "lead".
Line 46: Should this be Koch et al. (2020) instead of 2000?
Line 64: I believe 1960 should be "1960s".
Line 329: "reliably" instead of "reliable"
Figure 7 caption: misspelling of "charts" and "decomposition"

**Plan for revision:** *Thank you very much for these careful observations. All will be changed in the manuscript.*
**Changes:** *The misspellings were changed in the manuscript.*
* * *
Lines 101-114: What are the sources of uncertainty in the calculations of PET & ET from this model?

**Reply:** *The second reviewer stated their concern about an uncertainty of our ET estimate introduced by lowering the LAI over irrigated areas to rainfed conditions. By replacing the main component of crop and landcover characteristics over the irrigated area with a rainfed climatology LAI we will probably underestimate the rainfed ET baseline, thus potentially overestimating the net irrigation. Based on the reviewers' careful observations, we have reached the conclusion that we want to change our methodology in the manuscript, as we want to subtract rainfed ET from a managed scenario (this will result in higher ET baselines and lower net irrigation) rather than subtracting rainfed ET from a non-managed scenario (what we did in the original submission).*
**Plan for revision:** *We will run all the models without the LAI correction to estimate rainfed ET in a managed scenario. We expect this to decrease our net irrigation estimates. The two approaches will be briefly discussed in the discussion section of the revision.*
**Changes:** *We changed the method description on how the implementation of LAI climatologies in this study represent a managed scenario and thus deviate from the original framework by Koch et al. (2020) (lines: 107-111). This change in method has*

*resulted in an update of all irrigation results in the Abstract, Chapter 4 and Chapter 5
including an update of figures nr. 6 and 7.*
* * *
Line 101: Please include some more information on the ET calculation (e.g., an equation
or equations to show the ET calculation). Is there a soil water module involved in the
hydrologic model?

***Reply:*** *We agree that specifics about mHM's ET calculations may be of interest to some
readers. Since we are applying a well-documented mode, i.e. mHM, we have not included
any detailed information regarding the ET module in the manuscript. The scope of the
manuscript is the sole application of mHM to quantify the precision of the ET-based
irrigation approach, which, in our opinion, does not require the specifics on ET
calculations. In short, mHM calculate ET by reducing PET by fraction of roots and a soil
water stress factor, in our study, Fedde's equation from the defined soil layers.*
***Plan for revision:*** *We will add a few more lines on the ET and soil water modules in
mHM, but these will not be described by equations.*
***Changes:*** *We added further information on how ET is calculated in the mHM model setup
(lines: 103-105).*
* * *
Line 110: Please explain the choice of why FAO-56 PM was used to compute PET.

***Reply:*** Our choice of *FAO-56 PM is based on its documented ability to estimate PET for
use in irrigation management and studies evaluating FAO-56 PM against other PET
estimation methods.*

*Allen, R. G., Jensen, M. E., Wright, J. L., and Burman, R. D. (1989). "Operational estimates of evapotranspiration."
Agron. J., 81(4), 650-662.*

*Jensen, M.E., Burman, R.D. and Allen, R.G. (ed). 1990. Evapotranspiration and Irrigation Water Requirements. ASCE
Manuals and Reports on Engineering Practices No. 70., Am. Soc. Civil Engrs., New York, 360p*

*Martin, D. L., Gilley, J. R., Carmack, W. J., and Hardy, L. A. (1993). SCS methods for determining irrigation water
requirements." Management of irrigation and drainage systems: Integrated perspectives, R. G. Allen, ed., ASCE, New
York, 1031-1038*

***Plan for revision:*** *Will be further documented our choice in the manuscript and
substantiate our choice with the abovementioned references.*
***Changes:*** *We added a couple of lines and references on why the use of the FAO-56 PM
suits this study (lines: 119-121).*
* * *
Line 119: in the supplementary information, can you please provide the list of parameters
that were calibrated?

***Plan for revision:*** *List will be provided in supplementary material, with parameters, initial
values and parameter bounds.*
***Changes: A*** *table with end calibration parameters for all 16 calibrations and parameter
bounds has been provided in supplementary materials (table S1).*
* * *
Eq. (3): This definition of net irrigation assumes that 100% of applied irrigation is returned to the atmosphere via ET within the month it is applied. In some instances, this may not be the case. I suggest changing the title to "The precision of satellite-based net irrigation quantification in the Indus and Ganges basins"

**Reply:** *We agree that the title can be misleading without specifying that it is not total irrigation we are aiming at quantifying with this approach, but only the part related to evapotranspiration.*
**Plan for revision:** *We will change the title of the manuscript as suggested by the reviewer. And clarify what part of total irrigation our term net irrigation cover.*
**Changes:** *We changed the manuscript title by specifying that we are quantifying net irrigation and not actual irrigated water use. We added a further description on the term "net irrigation" in relation to the evapotranspiration approach in section 3.5 (222-224).*
* * *
Fig. 2b & 2d: please constrain the axes limits to the temporal domain of the study, which looks to be constrained by the ET data.

**Plan for revision:** *The axes limits will be constrained to the temporal domain of the study.*
**Changes:** *The figure 2 color and axes have been edited accordingly.*
* * *
Fig. 4: Please clean up this figure. Some labels on the left are cut off and the dots are awkwardly overlapping. The color scheme could also benefit from a change.

**Plan for revision:** *Figure will be edited.*
**Changes:** *The figure 4 color and legend have been edited accordingly.*
* * *
Line 265: To support that the baseline models can accurately simulate reference ET products, please include a time series comparison over the calibration period. A low MAE can result from large random biases, rather than accurate estimation of ET, thus the MAE is limited in information.

**Reply:** *The mean absolute error (MAE) is expected to capture large biases independent of their sign. We chose the MAE as objective function in the calibration to get the right magnitude of rainfed ET. Opposed, the mean error (ME) may be incentive to random biases if they are evenly distributed in positive and negative direction. However, we agree that a timeseries of the monthly MAE could be a good addition to the supplementary analysis.*
**Plan for revision:** *We will include time series comparisons from the calibration in supplementary material.*
**Changes:** *In this case, we decided not to include any figures in the supplementary materials but only here in the review answer to illustrate our first point made in the reply that a large random bias cannot explain a low MAE. We have chosen a random subset of 6 calibrations and plotted monthly ET timeseries from the calibration period (figure R1).*

[Figure]

*Figure R1: ET timeseries from 6 different calibrations.*

Figure 5: Why is the month of February chosen here? The spatial maps only show a single snap shot (in Feb.) of an instance that helps illustrate the points made in lines 272-285, but presentation of more data is needed to know that this snapshot is not a special case. Perhaps creating a boxplot for differences in ET (ref-baseline) of irrigated lands (across all periods of analysis) for all ET products would be beneficial here to show these results are generalizable, and maintain this snap shot to help illustrate the point through this single example.

*Reply: Yes, it is correct that the figure shows a single timestep that was intended to help to illustrate the overall approach. We would argue that the results from our irrigation ensemble estimates do confirm that this is not just a single case, but represent a good representation of the irrigation patterns. In the month of February, we expect the deviation between ET reference and ET baseline to be the largest, since irrigation activities peak during this month. The sole purpose of this Figure is to present our approach visually and not to quantify the ET differences. However, we agree that additional information on the differences in ET may be of interest to the readers.*
*Plan for revision: The differences in ET (ref-baseline) at monthly timescale will be prepared for all 15 models.*
*Changes: We decided not to create plots of differences in reference and baselines as these maps represent the irrigation ensemble, already shown in figure 7 in the manuscript as yearly seasonality for the four different climatic zones. Also, the fact that we see elevated evapotranspiration during the dry season in figure 2 (from the manuscript) (a mean climatology for a 10-year period) confirms that our references do capture the signal*

*of irrigation. We have therefore chosen to only add the following figure R2 to this review answer that show mean monthly ET between the three ET references for rainfed (green) and irrigated (blue) areas for the simulation period 2003-2012. The plot shows a significant difference between rainfed and irrigated areas during the dry season due to reoccurring seasonal irrigation.*

[Figure]

*Figure R2: Difference in mean monthly ET between the 3 ET references (FLUXCOM, NTSG, PML-v2) for rainfed (green) and irrigated (blue) areas for the simulation period 2003-2012.*
* * *
Figure 6: does this ensemble include ERA5-Land. I expect it does not because the prior plot just illustrated that irrigation is not present in this data set. Either way, please clarify this in the manuscript text. (Similar comment also applies to Figure 7). If ERA5-Land is included, shouldn't the estimated irrigation for that ensemble member be close to 0 resulting in a large reduction of the precision?

**Reply:** *You are right, ERA5-Land is not included in the ensemble.*
**Plan for revision:** *This will be clarified.*
**Changes:** *We added a line stating that ERA5-Land are excluded from the irrigation ensemble estimates (lines: 313-314).*
* * *
Line 290: Please note the ratio of the standard deviation to the mean irrigation to give a quantification of how large the ensemble uncertainty is relative to the magnitude of irrigation.

**Reply:** *This value was reported in line 309.*
**Plan for revision:** *Clarify*
**Changes:** *Nothing changed, values can be found in lines 355-356.*
* * *
Line 309: Why is precision much lower in arid regions?

**Reply:** *The precision is lower in the arid regions because of evident errors in the FLUXCOM and PML ET datasets. During the wet season both FLUXCOM and PML have almost no spatial variation in ET within the arid zone and so, ET is characterized by having a very high, uniform ET rate. NTSG and our hydrological models on the other hand show a*

*distinct spatial variation within the arid zone, which follows variability in vegetation, soil and climate. This is why the precision is lower in the arid zone when we compare irrigation estimates from the three ET products.*
**Plan for revision:** *We will clarify this in the manuscript.*
*Changes: We added some additional text further describing why the irrigation ensemble in the arid and semi-arid zones show high irrigation amounts and low precision. The added lines emphasize that these results are due to what we think are errors in FLUXCOM and PML products mainly occurring within the arid zone. (lines: 329-332)*
* * *
Lines 322-325: This statement claims lower precision is expected during the wet period because irrigation is lower during this time of year. However, Figure 7 estimates a peak in irrigation occurs during the wet part of the year in the arid and semi-arid regions. Please reconcile this.

**Reply:** *Thanks, this peak relates to the comment before. Errors in FLUXCOM and PML in the arid zone during the wet period yield high mean irrigation ensemble estimates. NTSG estimates of irrigation is much lower and therefore the precision is also very low.*
**Plan for revision:** *We will better explain this in the revised manuscript.*
*Changes: This question is related to the question above and therefore the action taken is also covered by lines: 329-332.*

[Reviewer comments in normal font; *Author replies in itialic*]

**REVIEWER 2**

**General comments:**

The net irrigation over two severely water-stressed basins (Indus and Ganges) is estimated by subtracting the satellite-based actual evapotranspiration (ET) from a baseline rainfed ET estimated through hydrological modeling. This study is a follow on to a previous study by Koch et al., 2020, with one significant enhancement of using an ensemble approach in which multiple precipitations and RS-based ET data are used to create an ensemble model simulation to estimate net irrigation and its uncertainty. The results are nicely presented and the manuscript is well-written. However, I believe the authors should distinguish between consumed irrigation by the crops (what is estimated in this study) and net irrigation water use which can be significantly higher than consumptive water use (based on irrigation efficiency). I am also concerned about replacing the observed LAI with the rainfed LAI climatology to calculate the rainfed component of ET. This can potentially lead to false baseline ET estimation by removing the irrigated crop characteristics. Details on these main concerns along with some other moderate and minor comments are provided below. Addressing all these comments I suggest the acceptance of the paper.

*Reply: We thank the reviewer for their overall positive and constructive feedback to our work. We will carefully address the point related to the irrigation definition and the LAI correction. Below, we outline how we consider responding to the issues pointed out by the reviewer in the revision and what changes were implemented*
* * *
Specific comments:

L44: There is an important distinction between the irrigation water consumed by crops and the net irrigation. ET is a measure of consumptive water use which is consumed irrigation water over the irrigated area. In many cases of flood or surface irrigation, a substantial portion of irrigation is lost to drainage (not consumed by the crops). This is especially important in your case studies where the irrigation efficiency is reported to be less than 52% on average (Simon et al., 2020). Please clearly mention in the manuscript that what is estimated here is consumed irrigation and not net irrigation. Simons, G. W. H., et al. "A novel method to quantify consumed fractions and non-consumptive use of irrigation water: Application to the Indus Basin Irrigation System of Pakistan." Agricultural Water Management 236 (2020): 106174.

*Reply: We agree with the reviewer, that there are multiple definitions of irrigation. In our case, by using the term "net irrigation" we are referring to the part of the total irrigation that has evapotranspired. By using an ET-based method our irrigation estimates contain both irrigation water consumed by crops (transpiration) and irrigation water evaporated from the soil and crop surface. Since we expect transpiration to dominate over evaporation from soil and water surfaces in such heavily farmed settings, we also expect that our definition of*

*net irrigation comes very close to the reviewer's definition of irrigation consumption. In our definition, we expect the net irrigation to be lower than actual applied irrigation. There can be re-infiltration leading to recharge or surface drainage. We thank the reviewer for sharing the relevant reference with us, which we will add to our discussion of the results.*
***Plan for revision***: *We will make a clarification of our and alternative ways to define irrigation in the manuscript and discuss that the actual total irrigation is expected to be higher than our net irrigation.*
***Changes:*** *We added a further description on the term "net irrigation" in relation to the evapotranspiration approach in section 3.5. Also we further added a couple of lines on why our term may not account for the irrigated water use at field scale, but overall may account for a substantial amount of the actual irrigated water use in a large complex system like Indus that are adapted to extensively reuse drainage water (lines: 222-224).*
* * *
The problem with replacing LAI over irrigated agriculture with climatology LAI over rainfed areas: The only place where the crop or land cover type is incorporated in the actual ET estimation in the hydrological model is in the downscaling of ET potential using the LAI data. Here I quote from a reference study (Demirel, et al., 2018) that is cited here for this part of the methodology: "The DSF (vegetation dynamic coefficient) is parametrized using spatiotemporal LAI component accounting for the effect of characteristics that separate the actual vegetation from a reference grass. These characteristics include specific landcover, albedo and aerodynamic resistance ...," here, you are replacing the main component of crop and landcover characteristics over the irrigated area with a rainfed climatology LAI which has different characteristics (land cover, crop type, albedo, etc.). This can lead to a false baseline ET estimate and consecutively net irrigation and can be a major source of error that needs proper attention and discussion in the manuscript. Please comment on the possible impact of this replacement on the final net irrigation estimation.

***Reply:*** *Thank you and this is indeed a relevant comment. Along these lines, also the first reviewer stated their concern about an introduced uncertainty of our ET estimate by lowering the LAI over irrigated areas to rainfed conditions. By replacing the main component of crop and landcover characteristics over the irrigated area with a rainfed climatology LAI we will probably underestimate the rainfed ET baseline, thus potentially overestimating the net irrigation. Based on the reviewers' careful observations, we have reached the conclusion that we want to change our methodology in the manuscript, as we want to subtract rainfed ET from a managed scenario (this will result in higher ET baselines, lower net irrigation) and not subtract rainfed ET from a non-managed scenario (what we are during now).*
***Plan for revision***: *We will run all the models without the LAI correction to estimate rainfed ET in a managed scenario. We expect this to decrease our net irrigation estimates. The two approaches will be briefly discussed in the discussion section of the revision.*
***Changes***: *We changed the method description on how the implementation of LAI climatologies in this study represent a managed scenario and thus deviate from the original framework by Koch et al. (2020) (lines: 107-111). This change in method has resulted in an update of all irrigation results in the Abstract, Chapter 4 and Chapter 5 including an update of figures nr. 6 and 7.*
* * *
L233: part of uncertainty can be attributed to the simplified model physics and the heterogeneity of land cover which is not mentioned in the manuscript. Please discuss these other sources of uncertainty in the manuscript as well.

**Reply:** *Thank you and it is correct that the model uncertainty and land cover parameter uncertainty are not addressed/quantified in our uncertainty analysis. The first could be addressed by including alterative model codes in the ensemble and the second on by using alternative LAI datasets. We believe that the precipitation input and ET dataset are most crucial for the irrigation quantification which we address in our submitted work and extending the analysis to included additional sources of uncertainty would go beyond our scope. Nevertheless, we agree with point raised and will add additional sources of uncertainty to our discussion.*
**Plan for revision:** *Discuss that we are not doing a complete uncertainty analysis, but also emphasize that we believe that we address to dominant sources.*
**Changes**: *We addressed this comment in section 3.6 regarding the uncertainty analysis. (lines: 259-260).*
* * *
L34: there is also a more recent study on irrigation mapping using SMAP-Sentinel1 1kmsoil moisture data that can be cited here: E. Jalilvand, R. Abolafia-Rosenzweig, M. Tajrishy and N. N. Das, "Evaluation of SMAP/Sentinel 1 High-Resolution Soil Moisture Data to Detect Irrigation Over Agricultural Domain," in IEEE Journal of Selected Topics in Applied Earth Observations and Remote Sensing, vol. 14, pp. 10733-10747, 2021, doi: 10.1109/JSTARS.2021.3119228.

**Reply:** *Thanks, we will take a careful look at the reference provided and it will be incorporated in the revised manuscript.*
**Changes**: *The article was used as reference (line: 35).*
* * *
L44: some other disadvantages of using ET that are not mentioned here: a. Limitation of ET estimation in cloudy weather situations b. The ET is an estimation of consumptive water use not irrigation

**Reply:** *We agree that RS-based ET cannot estimate the total amount of irrigation as some of the irrigation water potentially could leave the catchment through river discharge or recharge the groundwater. What the RS-based actual ET can provide is the amount of irrigation that evaporates and transpires. We are not too concerned with clouds since we aggregate the ET data to monthly timescale, which alleviates many of issues related to cloud coverage.*
**Plan for revision**: *Further describe the abovementioned disadvantages of using RS-based ET in the manuscript.*
**Changes**: *Other limitations were added regarding cloud cover (lines:46-47).*
* * *
L 62: There are many other studies on the satellite-based ET and consumptive water use estimation over the Indus and Ganges basins which can be referred to in the introduction or the discussion section of the paper.

Karimi, P., Bastiaanssen, W. G. M., Molden, D., and Cheema, M. J. M.: Basin-wide water accounting based on remote sensing data: an application for the Indus Basin, Hydrol. Earth Syst. Sci., 17, 2473–2486, https://doi.org/10.5194/hess-17-2473-2013, 2013.
Simons, G. W. H., et al. "A novel method to quantify consumed fractions and non-consumptive use of irrigation water: Application to the Indus Basin Irrigation System of Pakistan." Agricultural Water Management 236 (2020): 106174.
Peña-Arancibia, Jorge L., Joel P. Stewart, and John M. Kirby. "Water balance trends in irrigated canal commands and its implications for sustainable water management in Pakistan: Evidence from 1981 to 2012." Agricultural Water Management 245 (2021): 106648.

*Reply: Thank you and it is indeed possible that we have overlooked some key references for the Indus-Ganges basins.*
*Plan for revision: We will do another literature check and will add relevant references to the introduction and discussion sections.*
*Changes: References have been added in the introduction and discussion (line: 27) and (lines:222-224, 373-384).*
* * *
L113-114: a more recent Modis product version (v 061) was introduced at least a year ago (late 2020) and the research community is advised to use this product due to changes and improvements in the calibration approach. It is expected that the most recent product is used in a study that is going to be published in late 2022. It would be interesting if a test analysis were conducted using the v061 data and the differences were reported in the supplementary material.

*Reply: Before we selected the three ET products (FLUXCOM, PML and NTSG) we analyzed 11 different ET products by comparing their annual and monthly variability. The suggested MODIS product version (v 061) was part of this analysis but the dataset was not included in the final ensemble because the yearly ET for the basins were half of what other products estimated. We attribute this clear shortcoming to a large amount of NO DATA during the monsoon season (cloud cover) and odd ET rates of 0 mm/day for most of the two catchments during the dry period. We tried to calibrate the hydrological model against MODIS but were unable to calibrate the model to a satisfying level, which indicates that there is a substantial inconsistencies between precipitation, potential ET and MOD16 based actual ET.*
*Plan for revision: None.*
* * *
L127: is there any time dimension in the optimization conducted in this study or the optimization is only done in the space domain and on one image (Snapshot)? Can you comment on how different it would be if the optimization were conducted for each pixel separately and in time and why not time series based objective function is used in your optimization?

*Reply: The time dimension is part of both objective functions. MAE is calculated for each monthly timestep over the period of 2003 – 2007 and SPAEF is separated into wet and dry seasonal patterns. In this way, we believe that both the temporal and spatial performance*

*of the model is addressed. Since the first reviewer also raised a related point, we will add a plot of the monthly MAE to the supplementary material.*
***Plan for revision:*** *We will clarify the calibration design and prepare an additional plot for the supplementary material.*
*Changes: We have added some further descriptions to section 3.2 about the temporal and spatial dimensions of our calibration and validation setup. The figures in supplementary materials also helps to illustrate the overall strategy. (lines: 133-140).*
* * *
L207: net irrigation is a misleading phrase as explained in the major comment (1).

***Reply:*** *See reply to major comment (1)*
* * *
L290: please explain why the net irrigation precision is higher than the ensemble baseline rainfed ET.

***Reply:*** *The precision of the irrigation estimates is higher than the ET baselines because we are able to account for the large differences in rainfed ET by calibrating each hydrological model to the different ET products.*
***Plan for revision:*** *We will try to clarify this in the manuscript.*
*Changes: We added three lines in section 4.2 describing why the precision of the irrigation ensemble is higher and why this is important. (lines: 319-320).*
* * *
L378: I assume here the Author meant RS-based actual ET by the reference ET which is again misleading as the reference ET has a different meaning in the evapotranspiration community. I suggest using different terminology.

***Reply:*** *With reference ET we mean the ET dataset that was used as reference in the calibration, which is of course actual ET. We agree that the choice of terms can be misleading, but we have consistently used the term reference ET for the RS-based ET datasets used in calibration and for the subsequent irrigation quantification.*
***Plan for revision:*** *We will clearly define how reference ET should be understood in our study.*
*Changes: We added a clarification on our definition of ET reference to avoid ambiguity in relation to the use of this term. (line:225-226).*
* * *
L19: 25 mm/season is the average of two basins? Please explicitly mention

***Reply:*** *Yes, the 25 mm/season is the average dry season uncertainty for both Indus and Ganges*
***Plan for revision:*** *This will be clarified.*
*Changes: Were clarified by additional text. (line: 19).*

L19: I think an "of" is missing after "the robustness"
L46: Koch et al., 2020 …
L261: 16th …

**Plan for revision:** *Thanks, all three points will be corrected in the manuscript.*
**Changes***: We changed the misspellings accordingly.*
* * *
L265-266: this sentence is not clear to me please rephrase.

**Reply:** *Thank you. We wanted to state that the modelled ET baselines show the same yearly variability as the RS-based ET products they were calibrated against.*
**Plan for revision:** *We will clarify this in the manuscript.*
**Changes***: We have rephrased the sentence (lines: 291-293).*

---

## Author Response (AR2)

[Reviewer comments in normal font; *Author replies in itialic*]

**REVIEWER 1**

**Minor changes**

There are a few minor comments remaining that should be addressed prior to publication, the most important of which is that the manuscript still requires further grammatical editing throughout. I note some grammatical edits below which apply to the abstract for example.

Line 16: Should read "the Indus basin…"

Line 17: "period" should be "periods" and "the Ganges basin"

Line 20: "which is related to the…"

Line 361 & 366: Bodyko should by Budyko

**Reply:** *We agree the manuscript would benefit from grammatical editing.*
**Changes:** *We have changed the specific grammatical errors mentioned by the reviewer. Moreover, we throuroughly edited the manuscript and corrected remaining errors to our best ability.*

Line 26 starts with "today" then uses a 2010 citation to support the claim, consider removing "today"

**Changes:** *Today was deleted (line 26)*

Line 312: error for reference to fig 6e

**Changes:** *Error messages have been fixed (319)*

Line 29: please be consistent on using "percent" vs the symbol "%". I suggest using the symbol.

**Changes:** *"percentage" was changed to "%" symbol (line 29).*

The new supplementary figure (Figure S1) is interesting and useful. It can benefit from some edits & additional explanation in supplementary materials:
(i) Larger y label for time series
(ii) Noting that white areas in spatial maps indicates land areas not included in the specific calibration or validation data (I think)

Can you please also include more detail regarding exactly what the time series are representing? It would be useful to start with your hypothesis about what you expected differences between modeled and

observed cases to be for irrigation and rainfed areas. Then explain how your plots are either consistent or inconsistent with that.

Is the differential of these lines (obs – mod) for irrigated cropland supposed to be net irrigation? Please explain how negative irrigation is interpreted and handled in your model. Is there a systematic bias in the rainfed case for periods when modeled ET peaks? (why?)

**Changes:** *We have included a section describing Figure S1 and how it is used to validate the parameter transfer (lines 25-36). Figure S1 has also been edited. The comment about how to treat negative irrigation estimates has been added to the manuscript (line 243).*

[Reviewer comments in normal font; *Author replies in itialic*]

**REVIEWER 2**

**Minor changes**

L46-47: suggestion to make the new sentence more clear:
Moreover, cloud cover can cause incomplete data coverage that can affect the temporal resolution of the ET-based approach. To address this limitation, we have aggregated the original data into monthly estimates in this study.

**Reply:** *We agree the sentence could be reformulated.*
**Changes:** *The sentence was changed. (lines 46-47).*
* * *
2. L108-109: To provide more context on why you made these changes to your approach, I think it is better to explain what Koch et al., 2020 did in their study in more detail. Potentially, you can bring back some of the removed sentences from the last revision.

**Reply:** *We agree that more information on changes between the two studies is needed.*
**Changes:** *More information on the Koch et al. (2020) methodology was added (lines 110-113)*
* * *
3. L134: Adding "temporal" to the magnitude at line 134 made it complicated, I think the addition of "monthly" to the definition of MAE is enough and temporal can be removed.

**Changes:** *The word "temporal" was deleted (line 133).*
* * *
Figure 6) I suggest adding a legend for panel E

**Reply:** *Because panel E is supplementary to the actual irrigation results, we think that adding a legend would drag attention from the results.*
**Changes:** *We added further information in the caption to emphasize that the panel E data legend can be viewed in Figure 1 (line 786).*
* * *
Figure 7) You can add a border around the Indus & Ganges basin inset to make the gray area more visible

**Reply:** *We agree that the small maps could be more visible.*
**Changes:** *We added a darker shade to the other climate zones, making it easier for the reader to see the figure and navigate within the irrigated area (figure 7 was changed).*